# Medical negligence claims and the health and life satisfaction of Australian doctors: a prospective cohort analysis of the MABEL survey

Owen M Bradfield ,[1] Marie Bismark,[1] Anthony Scott,[2] Matthew Spittal[3]

¹Law and Public Health Unit, Centre for Health Policy, Melbourne School of Population and Global Health, The University of Melbourne, Melbourne, Victoria, Australia
²Melbourne Institute of Applied Economic and Social Research, The University of Melbourne, Melbourne, Victoria, Australia
³Centre for Mental Health, Melbourne School of Population and Global Health, The University of Melbourne, Melbourne, Victoria, Australia

**Correspondence to**
Owen M Bradfield;
owenmb@student.unimelb.edu.au

## ABSTRACT

**Objective** To assess the association between medical negligence claims and doctors' self-rated health and life satisfaction.

**Design** Prospective cohort study.

**Participants** Registered doctors practising in Australia who participated in waves 4 to 11 of the Medicine in Australia: Balancing Employment and Life (MABEL) longitudinal survey between 2011 and 2018.

**Primary and secondary outcome measures** Self-rated health and self-rated life satisfaction.

**Results** Of the 15 105 doctors in the study, 885 reported being named in a medical negligence claim. Fixed-effects linear regression analysis showed that both self-rated health and self-rated life satisfaction declined for all doctors over the course of the MABEL survey, with no association between wave and being sued. However, being sued was not associated with any additional declines in self-rated health (coef.=−0.02, 95% CI −0.06 to 0.02, p=0.39) or self-rated life satisfaction (coef.=−0.01, 95% CI −0.08 to 0.07, p=0.91) after controlling for a range of job factors. Instead, we found that working conditions and job satisfaction were the strongest predictors of self-rated health and self-rated life satisfaction in sued doctors. In analyses restricted to doctors who were sued, we observed no changes in self-rated health (p=0.99) or self-rated life satisfaction (p=0.59) in the years immediately following a claim.

**Conclusions** In contrast to prior overseas cross-sectional survey studies, we show that medical negligence claims do not adversely affect the well-being of doctors in Australia when adjusting for time trends and previously established covariates. This may be because: (1) prior studies failed to adequately address issues of causation and confounding; or (2) legal processes governing medical negligence claims in Australia cause less distress compared with those in other jurisdictions. Our findings suggest that the interaction between medical negligence claims and poor doctors' health is more complex than revealed through previous studies.

## INTRODUCTION

Medical negligence litigation allows patients who suffer injury or loss resulting from substandard medical care to seek redress through the legal system.[1] However, the

## STRENGTHS AND LIMITATIONS OF THIS STUDY

⇒ The key strength of this study is that we employed a longitudinal design and adjusted for relevant covariates when assessing whether experiencing a medical negligence claim is associated with a change in self-rated health and self-rated life satisfaction.

⇒ The use of seven years of cohort data, and the ability to adjust for demographic, vocational and psychosocial factors that contribute to poor doctor health, enabled clearer causal inferences to be drawn between medical negligence claims and psychosocial outcomes.

⇒ Doctors were lost to follow-up at the end of each wave, which may have resulted in a selection bias in that doctors with poorer health may have been less likely to remain in successive waves of the survey.

⇒ Data on exposure and outcome variables were based on self-report, as official statistics from courts or insurers on the number of doctors sued are not publicly available in Australia.

⇒ Only a small proportion of doctors participating in the survey were sued, which might have reduced our capacity to detect a statistically significant difference in self-rated health and life satisfaction between sued doctors and controls.

system has been criticised for being slow, expensive, inaccessible[2] and emotionally distressing for patients and doctors.[3] This is a societal problem,[4] as accumulating research shows that poor doctor well-being adversely affects doctors' clinical performance and decision-making, which can directly affect the quality of care that patients receive.[5]

Prior cross-sectional studies in the UK,[6] Ireland,[7] USA,[8] and Canada[9] have converged on the conclusion that medical negligence litigation adversely impacts doctors' health. For example, a recent self-report survey of nearly 8000 UK doctors found that doctors with a current or recent claim were approximately twice as likely as doctors without a claim to report suicidal ideation or

moderate-severe depression and anxiety. Another study showed that doctors who were sued experienced higher rates of depression, anxiety, post-traumatic stress disorder and suicidal ideation than doctors who were not.[10] Doctors who were sued identified the litigation process as their most stressful life experience: more so than divorce or the death of a spouse.[11] Even doctors who have not been sued identified the 'threat of litigation' as their most serious work-related stressor[12] and this was associated with doctors leaving the profession.[13]

The principal limitation of this prior body of research is that it almost entirely relies on cross-sectional surveys[14] that cannot easily define the temporal relationship between the causes and consequences of medical negligence claims. This limits the ability to infer a causal association. It also means, for instance, that it is not possible to reliably disentangle whether being sued increases the risk of poor health or whether poor health increases the risk of being sued, or both. It also limits the extent to which known risk factors for poor doctor health, including high job demands, low job control and low work-life balance can be considered as confounders or alternative explanations of any exposure-outcome association. These issues could be resolved using prospective cohort data because individuals can be followed over time, allowing observation of outcomes before and after a claim. It would also allow the impact of confounders on the exposure-outcome association to be observed.

In this study, we used the Medicine in Australia: Balancing Employment and Life (MABEL) cohort to assess the relationship between self-reported medical negligence claims and self-rated health and self-rated life satisfaction. We used a fixed-effects regression approach to control for unobserved stable individual factors as well as observed time-varying factors. Based on prior studies, we hypothesised that, compared with doctors who were not sued, doctors who were sued would experience poorer psychological outcomes when controlling for confounding factors (measures of job quality). We further hypothesised that among doctors who were sued, there would be a decline in psychological outcomes in the year of a claim and in the year or two following a claim when compared with their outcomes prior to being sued. Our primary outcome measure was self-rated health, and our secondary outcome measure was self-rated life satisfaction.

## METHODS
### Data source
MABEL is a longitudinal panel survey of doctors' working conditions, job satisfaction, work-life balance, health and life satisfaction. It comprised 11 annual waves that were collected between 2008 and 2018. The initial cohort was drawn from a national directory of 59 620 practicing doctors in Australia, with new medical graduates and newly registered overseas trained doctors invited to participate in subsequent waves. These new recruits replaced doctors

lost to follow-up and maintained the cross-sectional representativeness of the cohort. MABEL excluded doctors who had retired, were on leave or working overseas. MABEL was developed by researchers at the Melbourne Institute of Applied Economic and Social Research and Monash University, Melbourne, Australia. Copies of the survey instruments are publicly available[15] and a detailed description of the MABEL protocol and cohort has been published elsewhere.[16]

### Sample selection
We restricted our analysis to survey responses from waves 4 to 11 (2011 to 2018) because earlier waves did not include questions about medical negligence claims. We excluded doctors-in-training and hospital non-specialists because they are usually hospital employees. In Australia, hospitals are vicariously liable for the negligence of doctors within their employ.[17] This means that the hospital, rather than its employed doctors, would defend any medical negligence litigation.

### Variables of interest
Our primary outcome was self-rated health. The survey question was 'In general, would you say your health is: excellent, very good, good, fair, poor'. We recoded this variable so that higher scores indicated better health. This tool is a strong predictor of mortality[18] and has been used in other large health research surveys.[19] Our secondary outcome was self-rated life satisfaction. This was measured using a 10-point scale asking respondents to self-rate their life satisfaction from 1 ('completely dissatisfied') to 10 ('completely satisfied'). Higher scores are indicative of greater satisfaction. Surveys of life satisfaction have been shown to be stable and sensitive to changing life circumstances.[20]

Our key exposure variable was being a defendant in a medical negligence claim. This was constructed by combining responses to two questions: 'In the preceding 12 months have you been named in a medical negligence claim?' (yes or no) and 'how long ago did it happen?' (≤3 months, 4–6 months, 7–9 months, 10–12 months ago). Responses were coded 0 for all waves prior to being sued and 1 from the wave the respondent reported being sued onwards. Doctors who were not sued during the study period were coded 0 throughout.

To adjust for the potential confounding effect of job satisfaction, we constructed four variables which we included in our models: high job demands, low job control, poor social supports and work-life imbalance. These four variables were derived from the 'Job satisfaction' questions contained in the MABEL survey, which themselves were drawn from the Warr-Cook-Wall Job Satisfaction Scale,[21] and have been validated for use in the Australian medical workforce context.[22] Previous research has shown that higher scores on these four variables are associated with higher odds of poorer self-rated health.[23]

High job demands was measured using four items scored 0–4 (strongly disagree to strongly agree). These were 'It

is difficult to take time off when I want to', 'My patients have unrealistic expectations about how I can help them', 'Running my practice is stressful most of the time' and 'The majority of my patients have complex health and social problems'. Low job control was measured using five items: 'Freedom to choose your own method of working?' (scored from 0 'very dissatisfied' to 4 'very satisfied', reverse coded); 'Amount of variety in your work?' (scored from 0 'very dissatisfied' to 4 'very satisfied', reverse coded); 'Amount of responsibility you are given' (scored from 0 'very dissatisfied' to 4 'very satisfied', reverse coded); 'The hours I work are predictable' (scored from 0 'strongly disagree' to 4 'strongly agree') and 'I am restricted in my employment and/or the time and hours I work due to lack of available childcare' (scored from 0 'strongly disagree' to 4 'strongly agree'). Poor social supports were measured using three items scored 0 to 4 (strongly disagree to strongly agree). These were: 'I have a poor support network of other doctors like me'; 'I don't have many friends or family members in my current work location' and 'It is easy to pursue my hobbies and leisure interests in my current work location' (reverse coded). Work-life balance was measured using four items: 'The balance between my personal and professional commitments is about right' (scored from 0 'strongly disagree' to 4 'strongly agree'), 'My hours of work' (scored from 0 'very dissatisfied' to 4 'very satisfied'), 'I can take time off at short notice, for example if one of my children is ill or for a home emergency' (scored from 0 'strongly disagree' to 4 'strongly agree') and 'My colleagues understand the need for work-life balance' (scored from 0 'strongly disagree' to 4 'strongly agree'). High job demands, low job control, poor social support and work-life balance were all converted to z scores with means of 0 and standard deviations of 1.

We also adjusted for the potential confounding effect of age and hours worked per week. Age at each wave was coded into 5-year bands (<35, 35–39, 40–44, 45–49, 50–54, 55–59, 60–64, 65–69, ≥70 years). Working hours was coded into three categories based on the Australian Bureau of Statistics' definition of a standard full-time working week.[24] These were <35 hours per week, 35–45 hours per weeks and >45 hours per week.

We also had information on the following time-invariant variables that we set to their baseline values: sex (male, female); specialty (general practitioners, adult medicine physicians, surgeons, paediatricians, anaesthetists, pathologists and radiologists, emergency physicians, obstetricians and gynaecologists, ophthalmologists, psychiatrists, other); dependent children (collapsed into 'none' or 'one or more'); currently living with partner or spouse (yes or no) and geographical location (collapsed into three categories based on the five-category Australian Standard Geographical Classification.[25]

### Statistical analyses

We first described the characteristics of the cohort at baseline using counts and percentages. We then performed fixed-effects linear regression analyses to examine changes in self-rated health and self-rated life satisfaction. Fixed effects models are useful in the context of cohort data because they estimate the average within-person change in the outcome according to time-varying covariates entered into the model. Our key predictor was the binary variable coded 0 prior to being sued and 1 from the wave a doctor reported being sued onwards. Doctors who were never sued were coded 0 throughout. We adjusted for wave, hours worked, high job demands, low job control, poor social supports, work-life balance and age. We also performed *post hoc* tests to determine if there was an interaction (effect modification) between being sued and wave. Finally, to explore a possible temporal relationship between being sued and the outcomes, we undertook an analysis only among those doctors who reported being sued. Our key predictor was a variable that distinguished between the years prior to being sued, the year they were sued, 1 year after being sued, 2 years after being sued and so on. Thus, we used fixed effects methods to determine if self-rated health and satisfaction with life changed after doctors were sued, and if so, how long any change persisted for. Our models included the same set of covariates as the main models. All analyses were conducted using Stata 16.1 (Stata, College Station, Texas, USA).

### Patient and public involvement

It was not possible to involve patients or the public in the design, conduct, reporting or dissemination plans of our research. Results will be made available to MABEL participants at https://melbourneinstitute.unimelb.edu.au/mabel/results-and-publications/journal-articles.

## RESULTS

### Demographic details of survey participants

Between 2011 and 2018, 15 105 doctors were available for analysis. A total of 885 (5.90%) reported being sued at least once. The characteristics of doctors included for analysis are shown in table 1. They were predominantly male (55%), general practitioners (50%), working more than 45 hours per week (28%) and practicing in metropolitan areas (67%). The majority were living with their spouse or partner (77%) and had at least one dependent child (55%). The baseline mean score for self-rated health was 3.07 (SD=0.92, range 0–4), where the maximum score represented excellent health. The mean self-rated life satisfaction score was 7.42 (SD=1.62, range 1–10) where the maximum score represented complete satisfaction. Doctors were followed for between 1 and 8 waves, with a mean of 3.8 waves. One third of doctors (n=5847) completed five or more of the eight waves of data collection. Eighteen per cent of doctors reported being sued in the first wave (2011) and 7% reported being sued in the last wave (2018). The remaining 75% reported being sued in the intervening waves with the proportion sued for the first time declining in each wave.

**Table 1** Characteristics of doctors at baseline (n=15 105)

| Characteristic | N | % |
|---|---|---|
| Sex | | |
| Female | 6718 | 44.5 |
| Male | 8334 | 55.2 |
| Missing | 53 | 0.4 |
| Age group | | |
| <35 | 2387 | 15.8 |
| 35–39 | 2376 | 15.7 |
| 40–44 | 2117 | 14 |
| 45–49 | 1873 | 12.4 |
| 50–54 | 1832 | 12.1 |
| 55–59 | 1585 | 10.5 |
| 60–64 | 1105 | 7.3 |
| 65–69 | 702 | 4.6 |
| ≥70 | 634 | 4.2 |
| Missing | 494 | 3.3 |
| Specialty | | |
| General practitioner | 7539 | 49.9 |
| Adult medicine | 1760 | 11.7 |
| Surgery | 686 | 4.5 |
| Paediatrics | 427 | 2.8 |
| Anaesthesia | 1007 | 6.7 |
| Pathology and radiology | 437 | 2.9 |
| Emergency | 403 | 2.7 |
| Obstetrics and gynaecology | 393 | 2.6 |
| Ophthalmology | 181 | 1.2 |
| Psychiatry | 634 | 4.2 |
| Dermatology | 68 | 0.5 |
| Other | 612 | 4.1 |
| Missing | 958 | 6.3 |
| Work location | | |
| Metropolitan | 10 097 | 66.8 |
| Regional/Rural | 2775 | 18.4 |
| Remote | 1515 | 10 |
| Missing | 718 | 4.8 |
| Hours worked per week | | |
| <35 | 5081 | 33.6 |
| 35–45 | 5149 | 34.1 |
| >45 | 4182 | 27.7 |
| Missing | 693 | 4.6 |
| Dependent children | | |
| None | 5196 | 34.4 |
| One or more | 8318 | 55.1 |
| Missing | 1591 | 10.5 |
| Living with a spouse or partner | | |
| Yes | 11 623 | 76.9 |

**Table 1** Continued

| Characteristic | N | % |
|---|---|---|
| No | 2036 | 13.5 |
| Missing | 1446 | 9.6 |

### Association between being sued and self-rated health and life satisfaction

Multivariate fixed-effects linear regression analysis indicated that mean self-rated health declined each wave (coef.=−0.04, 95% CI −0.05 to −0.04, p<0.001) (table 2). There was no evidence that being sued was associated with any additional declines in self-rated health (coef.=−0.02, 95% CI −0.06 to 0.02, p=0.39). High job demands (coef.=−0.02 per 1 SD) increase in scores, 95% CI −0.03 to −0.01, p<0.001), low job control (coef.=−0.04 per SD increase, 95% CI −0.05 to −0.03, p<0.001) and poor social supports (coef.=−0.03 per SD increase, 95% CI −0.04 to −0.02, p<0.001) were all associated with lower self-rated health. Achieving work-life balance (coef.=0.04 per SD increase, 95% CI 0.03 to 0.05, p<0.001) was associated with higher self-rated health. There was no evidence that hours worked per week (p=0.22) or age (p=0.14) were associated with self-rated health. In a *post hoc* test, there was no evidence of an interaction between wave and being sued (p=0.13).

A similar set of findings emerged for self-rated life satisfaction. Mean life satisfaction declined during each wave of data collection (coef=−0.06, 95% CI −0.07 to −0.05, p<0.001) and being sued was not associated with any further decline (coef.=−0.01, 95% CI −0.08 to 0.07, p=0.91). Results for all other predictors were similar, as detailed in table 2, and in a *post hoc* test, there was no evidence of an interaction between wave and being sued (p=0.42).

### Temporal association between being sued and self-rated health and life satisfaction

Among doctors who had been sued, we found no evidence that self-rated health or self-rated life satisfaction changed in the years after a claim was made (table 3). Compared with a sued doctors' self-rated health in the years prior to a claim, there was no evidence that their health changed in the year of a claim (coef.=−0.01. 95% CI −0.07 to 0.06), the year after the claim (coef.=−0.01, 95% CI −0.09 to 0.07) or any of the other subsequent years. The same pattern of results was observed for life-satisfaction.

### DISCUSSION

Doctors' health has been described as a 'global health-care predicament', with emerging evidence that poor doctor well-being adversely affects healthcare quality and safety.[26] Doctors who are unwell take more time off work, leading to workforce understaffing, increased staff turnover and increased healthcare expenditure.[27] Similarly, studies have linked poor doctor health with

**Table 2** Fixed effects regression predicting self-rated health and satisfaction with life

| | Self-rated health (51 099 observations among 13 841 doctors) | | Satisfaction with life (51 119 observations among 13 821 doctors) | |
|---|---|---|---|---|
| | Coefficient (95% CI) | P value | Coefficient (95% CI) | P value |
| Medical negligence claim | | 0.39 | | 0.91 |
| No | Ref. | | Ref. | |
| Yes | −0.02 (−0.06 to 0.02) | | −0.01 (−0.08 to 0.07) | |
| Wave (per one wave increase) | −0.04 (−0.05 to −0.04) | <0.001 | −0.06 (−0.07 to −0.05) | <0.001 |
| Hour of work per week | | 0.22 | | 0.034 |
| <35 hours | Ref. | | Ref. | |
| 35–45 hours | 0.02 (−0.01 to 0.03) | | 0.03 (−0.01 to 0.06) | |
| >45 hours | 0.02 (−0.01 to 0.04) | | −0.01 (−0.06 to 0.03) | |
| High job demands (per 1 SD increase) | −0.02 (−0.03 to −0.01) | <0.001 | −0.11 (−0.12 to −0.09) | <0.001 |
| Low job control (per 1 SD increase) | −0.04 (−0.05 to −0.03) | <0.001 | −0.25 (−0.27 to −0.24) | <0.001 |
| Poor social supports (per 1 SD increase) | −0.03 (−0.04 to −0.02) | <0.001 | −0.15 (−0.16 to −0.13) | <0.001 |
| Work life balance (per 1 SD increase) | 0.04 (0.03 to 0.05) | <0.001 | 0.21 (0.19 to 0.22) | <0.001 |
| Age | | 0.14 | | <0.001 |
| ≤35 years | Ref. | | Ref. | |
| 35–39 years | −0.03 (−0.07 to 0.01) | | 0.01 (−0.07 to 0.09) | |
| 40–44 years | −0.05 (−0.10 to 0.01) | | −0.07 (−0.17 to 0.03) | |
| 45–49 years | −0.04 (−0.10 to 0.03) | | −0.18 (−0.30 to -0.06) | |
| 50–54 years | −0.03 (−0.10 to 0.05) | | −0.23 (−0.37 to −0.09) | |
| 55–59 years | −0.03 (−0.12 to 0.05) | | −0.21 (−0.37 to −0.05) | |
| 60–64 years | −0.01 (−0.11 to 0.09) | | −0.17 (−0.35 to 0.01) | |
| 65–69 years | 0.03 (−0.08 to 0.14) | | −0.06 (−0.26 to 0.15) | |
| ≥70 years | 0.04 (−0.09 to 0.17) | | 0.01 (−0.24 to 0.25) | |

suboptimal patient care and a doubling of the risk of medical errors.[28] Factors contributing to poor doctor health include: professional stressors (long working hours, shift work, workplace violence); a blame culture in medicine; fear and stigma of discussing health concerns with colleagues; and easy access to medications that leads to self-prescribing.[29] There is also mounting research suggesting that medicolegal claims and complaints may contribute to poor doctor health.[30]

Providing good patient care is central to doctors' professional identities. Doctors are notoriously self-critical and interpret allegations made against them during litigation as an assault on their professional competence and integrity,[31] which can lead to retraumatisation[32] and vocational disenchantment.[33] Litigation can also inflict financial and reputational damage,[34] while fear of future litigation can lead to defensive behaviours such as overinvestigation or avoidance of high-risk patients and procedures.[35] These effects may be compounded by legal advice that discourages doctors from speaking about their litigation experience with colleagues and peers for fear of compromising their claim or breaching confidential settlement terms.[36]

However, in contrast to previous research, our study did not find an association between medical negligence claims and self-rated health and self-rated life satisfaction.

There are several possible explanations for this. First, prior studies adopted a cross-sectional design, which means that causation and the impact of time and other confounders could not be verified. Second, prior studies often examined the association between doctor health and various types of medicolegal claims, such as complaints or regulatory investigations, rather than medical negligence litigation specifically. Complaints and regulatory investigations may affect doctors differently to, or more severely than, litigation. Third, the legal processes and frameworks governing medical negligence claims differ between jurisdictions. Those processes in Australia may cause less distress than processes overseas.

In Australia, tort law reforms enacted 20 years ago aimed at curtailing medical negligence litigation that may have positively impacted the litigation experience for doctors in Australia.[37] Following widespread concerns that the volume and cost of medical litigation was making medical indemnity insurance unaffordable and unavailable for many doctors,[38] reforms were introduced that included: (i) shortening time-limits within which proceedings may be initiated; (ii) limiting claims to 'significant' injuries; (iii) capping compensation payments and (iv) mandating mediation.[39] It is also compulsory for Australian doctors to have professional indemnity insurance.[40] As a result

**Table 3** Effect of time since claim on self-rated health or life satisfaction in doctors who were sued

| | Self-rated health (4615 observations among 882 doctors) | | Self-rated life satisfaction (4713 observations among 885 doctors) | |
|---|---|---|---|---|
| | Coefficient (95% CI) | P value | Coefficient (95% CI) | P value |
| Time since claim | | 0.99 | | 0.82 |
| Years prior to a claim | Ref. | | Ref. | |
| Year of claim | −0.01 (-0.07 to 0.06) | | −0.07 (-0.19 to 0.05) | |
| One year after claim | −0.01 (−0.09 to 0.07) | | −0.03 (−0.18 to 0.12) | |
| Two years after a claim | 0.01 (−0.08 to 0.10) | | −0.07 (−0.24 to 0.11) | |
| Three years after a claim | −0.01 (−0.11 to 0.10) | | −0.11 (−0.31 to 0.10) | |
| Four or more years after a claim | −0.02 (−0.15 to 0.11) | | −0.08 (−0.33 to 0.17) | |
| Wave (per one wave increase) | −0.05 (−0.07 to −0.03) | <0.001 | −0.05 (−0.08 to −0.02) | 0.006 |
| Hour of work per week | | 0.11 | | 0.009 |
| <35 hours | Ref. | | Ref. | |
| 35–45 hours | 0.02 (−0.04 to 0.08) | | 0.12 (0.01 to 0.24) | |
| >45 hours | 0.07 (−0.01 to 0.14) | | 0.21 (0.08 to 0.35) | |
| High job demands (per 1 SD increase) | −0.01 (−0.04 to 0.02) | 0.68 | −0.08 (−0.14 to −0.02) | 0.006 |
| Low job control (per 1 SD increase) | −0.05 (−0.07 to −0.02) | 0.001 | −0.34 (−0.39 to −0.29) | <0.001 |
| Poor social supports (per 1 SD increase) | −0.04 (−0.07 to −0.02) | 0.001 | −0.22 (−0.27 to −0.17) | <0.001 |
| Work life balance (per 1 SD increase) | 0.04 (0.01 to 0.07) | 0.003 | 0.24 (0.19 to 0.29) | <0.001 |
| Age | | 0.041 | | 0.012 |
| ≤35 years | Ref. | | Ref. | |
| 35–39 years | −0.21 (−0.38 to 0.04) | | 0.25 (−0.58 to 0.07) | |
| 40–44 years | −0.23 (−0.44 to 0.02) | | −0.49 (−0.89 to −0.09) | |
| 45–49 years | −0.10 (−0.34 to 0.15) | | −0.52 (−0.99 to −0.05) | |
| 50–54 years | −0.07 (−0.34 to 0.21) | | −0.54 (−1.07 to −0.01) | |
| 55–59 years | −0.03 (−0.34 to 0.28) | | −0.43 (−1.02 to 0.17) | |
| 60–64 years | 0.05 (−0.30 to 0.39) | | −0.39 (−1.05 to 0.27) | |
| 65–69 years | 0.07 (−0.31 to 0.45) | | −0.07 (−0.81 to 0.66) | |
| ≥70 years | 0.08 (−0.35 to 0.51) | | −0.01 (−0.84 to 0.82) | |

of these reforms, there are now fewer medical negligence claims against doctors. In our study, the proportion of doctors who reported being sued for the first time declined year on year. Of those medical negligence claims that are commenced, the overwhelming majority settle out of court on confidential terms. This may mean that Australian medical negligence claims are rarely subject to media scrutiny and are less likely to inflict financial or reputational damage on doctors. In addition, medical defence insurers and legal practitioners play a crucial role in supporting and educating sued doctors about the personal and professional impacts of legal processes.[41] Our results may also suggest that sued doctors in Australia are better supported professionally and personally, compared with overseas.

Nevertheless, despite the absence of a correlation between medical negligence claims and poor doctor health, our findings add weight to growing calls to improve doctors' health. Self-rated health and self-rated life-satisfaction declined on average throughout the duration of the study. Growing demands on doctors, associated with higher patient expectations, and increased administrative and regulatory requirements, may have contributed to this finding. We found that high job demands, low job control, poor social support and a lower levels of work-life balance were all associated with poorer self-rated health and life satisfaction. This is consistent with previous findings that showed an association between poor psychosocial working conditions and self-rated health in doctors.[42] It is likely that workplace factors have significantly contributed to declining doctor health over the seven years studied. This reinforces the pressing need for ongoing efforts to support doctors' health and well-being, particularly during the pandemic. Doctors are often ashamed to disclose to peers that they are unwell for fear of being judged. Being unable to share their experiences may exacerbate feelings of isolation. As a group, unwell doctors are often silent and invisible with few available avenues of peer support. This needs to change, as prior research shows that doctors enjoy

better psychological well-being when supported by family, colleagues or employers.[43]

Our study had several limitations. First, a number of doctors were lost to follow-up at the end of each wave, which may have resulted in a selection bias in that doctors with poorer health may have been less likely to remain in successive waves of the survey. Second, data in relation to exposure to a medical negligence claim and the primary and secondary health and life satisfaction outcomes were self-reported, as official statistics from courts or insurers on the number of doctors sued is not publicly available in Australia. Third, as only a small proportion of doctors participating in the survey were sued, we were unable to detect a statistically significant difference in self-rated health and life satisfaction between sued doctors and controls. In a *post hoc* power calculation, we estimated that we had 13% power to detect the observed difference of −0.02 between sued and non-sued doctors on self-rated health. However, this difference is very close to zero, and the explanation that there is no association between being sued and self-rated health or life-satisfaction after adjustment for time seems more likely than the explanation that the study lacked power. This is because an indication of lack of power would be a large effect size with wide confidence intervals that include the null value, whereas we observed small effect sizes close to zero. A substantially larger sample of doctors who had been sued would be required to detect a difference between groups of this magnitude. Despite these limitations, the key strengths of this study were: its large sample size; the prospective cohort study design that enabled us to draw stronger causal inferences than previous studies; the assessment of a wide range of demographic, vocational and psychosocial covariables and controlling for time-invariant bias within and between persons.

While there are reports of doctors who have died by suicide in the context of medical regulatory investigations,[44] our large longitudinal analysis of doctors in Australia found no association between medical negligence claims and poor doctors' health. This may be because medical negligence claims have less impact on doctors compared with regulatory complaints or investigations. It may also be because any adverse impact of claims on doctors' health is short-lived and does not translate into lower self-rated health or life satisfaction scores 12 months later. Instead, we found a significant association between workplace factors and doctors' health, suggesting that workplace health and safety reforms, rather than further tort law reforms, ought to be the priority for ongoing work in improving the health and well-being of doctors and thereby benefitting the patients they serve.

**Acknowledgements** This research used data from the MABEL longitudinal survey of doctors. MABEL was developed by the Melbourne Institute of Applied Economic and Social Research and Monash University, Melbourne, Australia. Funding for MABEL was provided by the National Health and Medical Research Council (2007 to 2016: 454799 and 1019605); the Australian Government Department of Health and Ageing (2008); Health Workforce Australia (2013) and in 2017 The University of Melbourne, Medibank Better Health Foundation, the NSW Ministry of Health and the Victorian Department of Health and Human Services. In 2018, funding was provided by the Australian Government Department of Health.

**Contributors** OMB and MS developed the initial idea and methodology for the study and undertook the statistical analyses. They are also jointly responsible for the overall data as guarantors. AS provided technical expertise about the MABEL protocol and variables. MB assisted with the interpretation of results from a patient safety and doctors' health perspective. MB and AS revised the draft critically for important intellectual content. All authors have given final approval for the article to be published.

**Funding** OMB was funded by the Australian Government on a 2020 Melbourne Research Scholarship and is a 2020 Fulbright Future Scholar. MB was funded by the Australian Government on a National Health and Medical Research Council Investigator Grant (APP1195984). MS was funded by the Australian Government on an Australian Research Council Future Fellowship (FT180100075).

**Competing interests** None declared.

**Patient and public involvement** Patients and/or the public were not involved in the design, or conduct, or reporting, or dissemination plans of this research.

**Patient consent for publication** Not applicable.

**Ethics approval** The MABEL survey was approved by The University of Melbourne Faculty of Business and Economics Human Ethics Advisory Group (Ref. 0709559) and the Monash University Standing Committee on Ethics in Research Involving Humans (Ref. CF07/1102 -2007000291). Our study was approved by the Melbourne School of Population and Global Health Human Ethics Advisory Group (Ref. 1956096). The Melbourne Institute: Applied Economic and Social Research granted access to deidentified MABEL survey responses. Participant consent for publication was not required. Participants gave informed consent to participate in the study before taking part.

**Provenance and peer review** Not commissioned; externally peer reviewed.

**Data availability statement** Data may be obtained from a third party and are not publicly available. Deidentified MABEL survey data are available on application to the Australian Data Archive.

**ORCID iD**
Owen M Bradfield http://orcid.org/0000-0002-8955-7432

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
