## [Reviewer comments · BMJ Open]

ARTICLE DETAILS

TITLE (PROVISIONAL)	Medical negligence claims and the health and life satisfaction of Australian doctors: a prospective cohort analysis of the MABEL survey.
AUTHORS	Bradfield, Owen; Bismark, Marie; Scott, Anthony; Spittal, Matthew

VERSION 1 – REVIEW

REVIEWER	Sheikhazadi, Ardeshir Tehran University of Medical Sciences, Legal Medicine
REVIEW RETURNED	19-Jan-2022

GENERAL COMMENTS	I have published an article in this domain as "A survey of sued physicians' self-reported reactions to malpractice litigation in Iran" in JGIM 16 (6) 2009 (301-306). We saw the threat of, or real, legal process can cause psychological changes. But as you know there is much differences between legal process in different countries and a stressful process in some countries can produce psychological changes.
--

REVIEWER	Khan, Hanna Universiti Putra Malaysia
REVIEW RETURNED	22-Jan-2022

GENERAL COMMENTS	This is a great paper. Yes, this area of research has yet to be covered by other authors.
---

REVIEWER	Geraedts, Max University of Marburg, Institute for Health Services Research and Clinical Epidemiology
REVIEW RETURNED	25-Mar-2022

GENERAL COMMENTS	The authors analyze the association between medical negligence claims and doctors' self-rated health and life satisfaction. For this purpose, they use survey data from the eight-year period 2011-2018 of the survey "Medicine in Australia: Balancing Employment and Life (MABEL)," which has been conducted annually in Australia for several years. As a result, the authors find no association between medical negligence claims and doctors' self-rated health and life satisfaction, in contrast to several cross-sectional studies from other countries. The rationale of the study and the hypothesis derived from it - medical negligence claims have a negative impact on physicians' health as well as on their life satisfaction - are well derived. The methodological approach is adequately chosen. The results are
---

	clearly presented and the discussion, including strengths and limitations, and conclusions are appropriate. The manuscript should be supplemented in a few places:  - the authors do not justify why exactly the items listed were used when asking about the constructs “High job demands, low job control, poor social supports, and work-life imbalance” from the MABEL-Questionnaire and from which survey instruments these items were taken. If a psychometric test of the questionnaire is available that proves the fit of the items to the constructs, this should be cited. Otherwise, this would be cited as a limitation. - The authors state that a small number of sued physicians might have led to the fact that an existing association could not be detected. Therefore, a power analysis should be added to allow the reader to assess the significance of the nonsignificant associations. - Last, a better estimate of self-reported data on the frequency of lawsuits could be obtained by comparing the data in the surveys with official statistics from the courts or insurers on the number of physicians sued.
--	--

VERSION 1 – AUTHOR RESPONSE

Comments from Dr Ardeshir Sheikhzardi (Reviewer 1)

1. I have published an article in this domain as "A survey of sued physicians' self-reported reactions to malpractice litigation in Iran" in *jflm* 16 (6) 2009 (301-306). We saw the threat of, or real, legal process can cause psychological changes. But as you know there is much differences between legal process in different countries and a stressful process in some countries can produce psychological changes.

Thank you for drawing our attention to this important prior work. We have already noted in our manuscript that “legal processes and frameworks governing medical negligence claims differ between jurisdictions” and that one reason why our results differ to prior overseas findings is that “processes in Australia may cause less distress than processes overseas”.

Comments from Dr Hanna Khan (Reviewer 2)

1. This is a great paper. Yes, this area of research has yet to be covered by other authors.

Thank you for your support for our manuscript.

Comments from Dr Max Geraedts (Reviewer 3)

1. The authors do not justify why exactly the items listed were used when asking about the constructs “High job demands, low job control, poor social supports, and work-life imbalance” from the MABEL-Questionnaire and from which survey instruments these items were taken. If a psychometric test of the questionnaire is available that proves the fit of the items to the constructs, this should be cited. Otherwise, this would be cited as a limitation.

The MABEL survey contained a series of “Job satisfaction” questions that were drawn from the Warr-Cook-Wall Job Satisfaction Scale, a modified version of which was validated for use in the Australian clinical medical workforce. The four variables that we constructed (“High job demands, low job control, poor social supports, and work-life imbalance”) were theoretically derived from this existing validated measure of job satisfaction, but were not themselves formally validated. They have nevertheless been employed by others in assessing the relationship between working conditions and self-rated health among Australian doctors.

We have now revised the manuscript to include this information. The text reads (page 7):

To adjust for the potential confounding effect of job satisfaction, we constructed four variables which we included in our models: high job demands, low job control, poor social supports, and work-life imbalance. These four variables were derived from the “Job satisfaction” questions contained in the MABEL survey, which themselves were drawn from the Warr-Cook-Wall Job Satisfaction Scale, and have been validated for use in the Australian medical workforce context. Previous research has shown that higher scores on these four variables are associated with higher odds of poorer self-rated health.

2. The authors state that a small number of sued physicians might have led to the fact that an existing association could not be detected. Therefore, a power analysis should be added to allow the reader to assess the significance of the nonsignificant associations.

Our comment about a lack of power intended to signal that this could be an explanation but not one we considered likely. An indicator of a lack of power would be a large effect size with wide confidence intervals crossing zero. That is not what we observed here. Rather, we saw effect sizes that were very close to zero on scales that ran from 0 to 4 (self-rated health) or 1 to 10 (life satisfaction). This pattern is more consistent with there being no association between being sued and self-rated health or life satisfaction. We have now made this point clearer. To respond to the comment, we have done a power calculation, and this shows that we have around 13% power to detect a difference of -0.02 between exposure groups on self-rated health. The text now reads (page 13):

Third, as only a small proportion of doctors participating in the survey were sued, we were unable to detect a statistically significant difference in self-rated health and life satisfaction between sued doctors and controls. In a post hoc power calculation, we estimated that we had 13% power to detect the observed difference of -0.02 between sued and non-sued doctors on self-rated health. However, this difference is very close to zero, and the explanation that there is no association between being sued and self-rated health or life-satisfaction after adjustment for time seems more likely than the explanation that the study lacked power. This is because an indication of lack of power would be a large effect size with wide confidence intervals that included the null value; whereas we observed small effect sizes close to zero. A substantially larger sample of doctors who had been sued would be required to detect a difference between groups of this magnitude.

3. Last, a better estimate of self-reported data on the frequency of lawsuits could be obtained by comparing the data in the surveys with official statistics from the courts or insurers on the number of physicians sued.

Thank you for this comment. We had thought of this, but this court information is not publicly available in Australia and insurers do not release this information. Even the US, which has one of the best systems for capturing claims through the National Practitioner Data Bank does not include claims not

substantiated. We have added the following text to the 'Discussion' and 'Strengths and Limitations' sections our manuscript to clarify this:

"...as official statistics from courts or insurers on the number of doctors sued is not publicly available in Australia".

VERSION 2 – REVIEW

REVIEWER	Geraedts, Max University of Marburg, Institute for Health Services Research and Clinical Epidemiology
REVIEW RETURNED	22-Apr-2022
GENERAL COMMENTS	The authors have sufficiently addressed the criticisms I raised during the initial review in the revision of the article. I therefore advocate acceptance of the article in its present form.